# The Effectiveness of Group and Individual Training in Emotional Freedom Techniques for Patients in Remission from Melanoma: A Randomized Controlled Trial

**DOI:** 10.3390/healthcare13121420

**Published:** 2025-06-13

**Authors:** Aneta Lazarov, Dawson Church, Noa Shidlo, Yael Benyamini

**Affiliations:** 1Independent Researcher, Skin Cancer Clinic, Herzliya 4678525, Israel; anetalazarov@gmail.com; 2National Institute for Integrative Healthcare, Petaluma, CA 94953, USA; dawsonchurch@gmail.com; 3Bob Shapell School of Social Work, Tel Aviv University, Tel Aviv 6139001, Israel

**Keywords:** melanoma, skin neoplasms, cancer survivors, mind–body therapies, psychological wellbeing, stress, psychological, emotional freedom techniques

## Abstract

**Background/Objectives:** A history of cancer has been linked to stress and concerns about its recurrence. We aimed to test the benefits of an evidence-based self-help stress reduction method, the Clinical Emotional Freedom Technique (EFT), in survivors of cutaneous melanoma, and to contrast its effects on wellbeing and perceptions of cancer recurrence when delivered in a group versus individual instruction setting. **Methods:** This study was preregistered at clinicaltrials.gov (NCT05421988, 3 April 2022). Fifty-three patients aged 18 and above, diagnosed with melanoma (stage T1a–T2a) at least 6 months prior, and not in active treatment were recruited from a private skin cancer clinic. After consent, all participants were randomized in one step into three condition groups: Group EFT (G-EFT; *n* = 16), Individual EFT (I-EFT; *n* = 18), and a waiting-list control condition (CC; *n* = 19). G-EFT and I-EFT participants attended weekly treatment sessions for four weeks. Perceptions of cancer recurrence and wellbeing measures were obtained pre- and post-intervention and at three-months follow-up using online questionnaires. Subjective units of distress (SUDs) were recorded by the EFT instructor at the beginning and end of each session. **Results:** Two-way repeated measures ANOVAs revealed significant improvements from pre- to post-intervention in both EFT conditions in terms of participants’ understanding of how to prevent recurrence and in their spiritual wellbeing. No statistically significant effects were found for fear of recurrence, recurrence perceptions, and affect. Significant decreases in SUD scores were observed in both EFT conditions. Over 80% of the experimental conditions’ participants reported positive changes and satisfaction. **Conclusions:** The findings provide support for offering EFT instruction as a non-pharmacological and noninvasive self-help method to ameliorate the stress of cancer diagnosis and treatment, and for its similar effectiveness in either a group or individual format.

## 1. Introduction

Skin cancers are the most commonly diagnosed group of cancers, with more than 1.5 million new cases estimated in 2020 worldwide; of these, malignant melanomas were estimated to account for approximately 325,000 cases [1]. Although the risk of melanoma generally increases with age, melanoma is already among the most common cancers in young adults [2]. Melanoma patients’ stress levels may be increased by the need to cope with disease management as well as with fear of cancer recurrence, medical follow-up regimens, self-examination, and adherence to sun protection measures.

The new integrative medicine paradigm in cancer patients’ care, including melanoma, includes a focus on the patients’ perception of wellness as well as their health-related quality of life (HRQOL) [3,4]. A review from 2009 showed that HRQOL is often affected by melanoma and deteriorates following melanoma diagnosis, with one third of the patients experiencing considerable levels of distress at the time of diagnosis and following treatments [3]. With the advancement of the disease, further HRQOL impairment is associated with patients’ health status and psychological characteristics. HRQOL and various psychosocial variables have been described as independent predictors of time to relapse and survival in patients with early-stage melanoma [5] and of survival in patients with advanced melanoma [6].

Several studies have investigated the role of stress in cancer appearance and progression [7,8]. The association between adult and childhood trauma and the risk of developing cancer is particularly relevant. Adverse childhood events have been found to increase the odds of any type of cancer [9,10] and also specifically of receiving a melanoma diagnosis [11]. Stressful events in childhood have been found to increase the odds of melanoma occurrence both directly and indirectly through the more negative effects of stressful events in adulthood [12]. A recent large study of adult skin cancer survivors reported that 56% of the participants experienced at least one adverse childhood experience (ACE) [13]. In that study, skin cancer survivors with a history of ACEs reported significantly poorer physical health and mental health compared to those without ACEs. They also experienced higher levels of lifestyle impairment related to health. In another study of patients with Basal Cell Carcinoma (BCC), another type of skin cancer, those who had both experienced a significant life stressor in the previous year and were emotionally maltreated by their parents as children evidenced diminished local immune response to their BCC tumor [14].

A review of 44 studies reported clinically relevant levels of anxiety and depression in a substantial portion of melanoma patients [15]. Melanoma patients showed an altered immune response to stress, compared to healthy individuals [16]. Higher rates of anxiety (25%) and psychological distress symptoms (44%) were identified in early-stage melanoma patients [17]. These and other psychosocial factors have been found to be related to changes over time in HRQOL in patients with melanoma [18]. Therefore, relieving stress and developing positive coping strategies is of particular importance in melanoma patients, since they remain at risk of disease recurrence and progression for many years after diagnosis.

The current study aimed to test the benefits of an evidence-based self-help method, Clinical Emotional Freedom Technique (Clinical EFT), for survivors of cutaneous melanoma. Delivery in a group setting was compared to individual instruction and a control condition to determine if the method could be beneficial as a stress reduction and coping strategy to reduce distress and enhance wellbeing.

Clinical EFT has been validated in research studies that meet the standards reflected in the recommendations of the American Psychological Association (APA) Division 12 Task Force on Empirically Validated Treatments [19,20]. EFT combines elements of exposure and cognitive therapies with the stimulation of acupuncture points with the fingertips (acupressure).

The exposure component of EFT guides clients to focus repeatedly on traumatic life events. However, they contextualize these distressing memories within a cognitive framework of self-acceptance. Simultaneously, somatic stimulation is produced using fingertip percussion on a series of acupuncture points. fMRI studies show that this combination of cognitive processing and fingertip tapping rapidly reduces arousal of the fear-processing structures in the brain [21,22].

The most extensive systematic review of the literature on the use of Clinical EFT [19] identified 56 randomized controlled trials (RCTs). The use of EFT in primary care settings has been found to be safe, rapid, reliable, and effective, whether delivered in-person or virtually, and symptom improvements persist over time. Treatment was associated with measurable biological effects in the dimensions of gene expression, brain synchrony, hormonal synthesis, and a wide range of biomarkers. The review confirmed Clinical EFT to be efficacious for a range of psychological and physiological conditions: (a) psychological conditions such as anxiety, depression, phobias, and post-traumatic stress disorder (PTSD); (b) physiological issues such as pain, insomnia, and autoimmune conditions; (c) professional and sports performance; and (d) biological markers of stress. Eight meta-analyses evaluating the effect of EFT treatment that were also identified in this review have found it to be “moderate” to “large” [19].

Another systematic review reported that as of 2023, peer-reviewed research on integrating the manual stimulation of acupuncture points into psychotherapy included 28 systematic reviews or meta-analyses; 125 clinical trials; 24 case studies; 26 reports detailing systematic observations; 17 mixed-method clinical trials with an acupressure tapping component; and 88 articles discussing clinical procedures, theory, mechanisms, or related matters [23]. In addition, more than 90 additional clinical trials investigating EFT or close variations have been published in non-English language journals [24].

An additional review assessed a database of peer-reviewed articles on energy psychology and focused on 245 clinical trials, meta-analyses, systematic evaluations, and theory pieces examining protocols that include tapping on acupuncture points [25]. This review described six premises about the efficacy of this method and its speed, durability, and physiological effects, which had sufficient empirical support to serve in delineating and making claims about the approach. These claims include that acupoint tapping protocols (a) are effective in treating a range of clinical conditions, (b) are rapid compared to conventional psychological treatments, (c) lead to durable benefits, (d) produce changes in biomarkers that corroborate the subjective assessments of clients, (e) are a critical ingredient of the demonstrated clinical effects, and (f) send signals that can increase or decrease arousal in specific areas of the brain [25].

The standard EFT protocol has a client recall an emotionally triggering memory. While focusing on the issue, the client rates their degree of emotional distress on a scale called subjective units of distress (SUDs) [26,27]. SUD scores can range from zero (indicating no distress or neutral) to 10 (indicating the highest possible level of distress). The client is instructed to tap on the side of the hand using a “Setup Statement” that involves naming the distressing event while pairing the memory with a statement of self-acceptance.

This procedure is designed to combine exposure (vivid recall of the traumatic memory) with cognitive framing (self-acceptance statement). The tapping continues on a sequence of eight selected acupuncture points (acupoints), while the client repeats a “Reminder Phrase” designed to maintain focus on the emotionally triggering event. The procedure is repeated till the client reports a low SUD score. Once the SUD score around a traumatic event approaches zero, attention is directed to the next emotional memory. It takes about a minute to tap on all 13 points identified in Clinical EFT, and this is referred to as a “round” of tapping. Several rounds of EFT might be required to bring the SUD score to zero or a low number. As the client’s focus of attention shifts, often to associated emotional events, the Setup Statement and Reminder Phrase are adjusted.

The acupressure component of EFT (namely tapping on the acupuncture point) has been shown to be an essential ingredient in the method’s efficacy and not merely a placebo. Six individual studies and a meta-analysis have found that tapping on acupoints is more effective than tapping on sham points or other active controls [28]. The somatic stimulation of acupressure thus contributes to the observed effects of Clinical EFT, in addition to the established psychological components of exposure and cognitive framing.

Several studies have investigated the effects of EFT on physiological dimensions including somatization, pain, physical symptoms, weight loss, insomnia, gene expression, autoimmune conditions, hormones, and cravings. Many physical symptoms have responded favorably to EFT treatment, such as frozen shoulder [29], fibromyalgia [30], pain [31,32,33,34], obesity [35,36], and traumatic brain injury [37].

Recent studies have demonstrated positive changes in physiological markers of health following acupoint tapping sessions and measured decreases in blood pressure, resting heart rate, and cortisol levels. These correlate with reductions in anxiety, depression, PTSD, chronic pain, and cravings [38]. All these studies indicate that EFT is effective in reducing somatic symptoms in a variety of populations and settings as a fast-acting, non-pharmacological, and evidence-based self-help method [19].

The application of EFT is also associated with the heightened expression of genes that influence immunity, inflammation, cancer suppression, cell metabolism, and the stress response [39,40]. Clinical EFT has also been found to attenuate the synthesis of stress hormones [41,42], normalize brainwave patterns [43], disrupt stress-related stimulus–response neural firing [44], regulate brain centers associated with fear [21,22], produce epigenetic changes in microRNAs associated with stress [45], and regulate biomarkers such as heart rate, immunity, and blood pressure [38].

The effect of EFT treatment has been measured in a number of studies of cancer patients. In an examination of menopausal symptoms, fatigue, and pain in women with breast cancer undergoing treatment with tamoxifen and aromatase inhibitors, all symptoms were significantly reduced [46]. Another study reported a reduction in stress, anxiety, and depression in cancer patients following an EFT intervention [47]. Women who underwent breast cancer surgery experienced significant improvements in sleep quality and happiness immediately and one month after treatment with EFT [48]. Additionally, a randomized controlled trial found that the application of EFT was associated with improvements in cancer-related cognitive impairment and subjective cognitive complaints [49]. It identified a statistically significant reduction in cognitive impairment scores and in distress, depressive symptoms, and fatigue, as well as an improvement in quality of life (QOL). In a trial that included a spiritual component, EFT was more effective in managing cancer pain than analgesic therapy alone [50].

In the current study, we adopted the multidimensional approach of Bonacchi et al. [51], which draws from the WHO definition of health as “a state of complete physical, mental and social wellbeing, not merely the absence of disease or infirmity” ([52] p. 1). Bonacchi et al. [51] added a fourth aspect, that of spiritual wellbeing. A review of 34 studies showed that spirituality (even more than religiosity) is associated with resilience in a variety of populations. These include healthy individuals as well as those diagnosed with chronic diseases. A recent review of 371 articles on spirituality in serious illness, assessed by multidisciplinary Delphi panels of experts, affirmed that spirituality is an essential component of person-centered care [53].

Another type of outcome we examined is the cognitive framing of illness, specifically the perception of cancer recurrence. A meta-analysis covering many disease populations, including cancer patients, found associations between their subjective perceptions of disease and physical and mental health outcomes [54]. Such associations between illness perceptions and quality of life have also been found among dermatological patients [55]. The concept of illness perceptions or illness representations is based on the commonsense model of self-regulation developed by Leventhal and colleagues [56,57]. According to this model, individuals react to a diagnosis based on their conceptualized idea of the disease, which is developed both cognitively and emotionally through a collection of personal experiences and exposure to information from external sources. This model emphasizes that illness perception and coping procedures are related, and both have a critical impact on patients’ adherence to treatment and sense of wellbeing.

The hypothesized associations between illness perceptions and coping as well as those between illness perceptions and outcomes have been extensively documented in many chronic diseases and also specifically among cancer patients (see meta-analysis; [58]). Optimistic illness perceptions were found to be associated with better health-related quality of life and survival, even if patient perceptions were unrealistic based on their prognoses [59]. Patients with pessimistic illness perceptions had the worst outcomes.

Breast cancer patients with pessimistic views of their diagnosis and prognosis and with a fear of progression have poorer physical and mental health outcomes [60]. These views are associated with helplessness, anxiety, depression [61], low medical compliance [62], and low QOL [63]. Changes over time in cancer patients’ illness perceptions and depressive symptoms were related to their coping levels, suggesting that fostering a sense of control and the adoption of positive coping strategies can promote psychological health [64]. This is supported by studies showing that patients who describe their coping efforts as attempts to take control over their cancer use more proactive strategies and are more confident of being cured than those who do not [65]. Maintaining a positive attitude and adopting a healthy lifestyle contributed to cancer patients’ sense of control over the course of their disease as well as favorable outcomes [66].

Most importantly for cancer survivors, illness perceptions are related to fear of recurrence [67,68], even when adjusted for general dispositional optimism [69]. A review of 18 studies revealed that representations of the initial cancer are distinct from representations of the risk of recurrence, each showing different relationships with worry about recurrence and behaviors that are protective of health [70].

Social support is another important factor in the progression and severity of various diseases, including cancer [71,72], as well as in survivors’ quality of life [73]. The latter study also showed that support was impaired by survivors’ uncertainty regarding their disease. Social support was found to be related to illness perceptions among patients with skin tumors [74] and to mediate the associations between illness perceptions and wellbeing in breast cancer patients [75]. We therefore structured the current study to compare EFT instruction and practice in a small group setting with individual sessions, to determine if groups add an element of peer social support.

A promising approach to empowering individuals and improving their wellbeing is through teaching them to adopt stress management techniques [76]. As cited earlier, Clinical EFT is effective in treating anxiety, depression, and PTSD, among other psychological conditions. In the current study we aimed to test its effectiveness in increasing wellbeing and reducing negative emotional symptoms in general, and reducing fear of recurrence in particular, among individuals previously diagnosed with melanoma. Specifically, we conducted an RCT aiming to achieve the following:To assess the effect of the instruction and practice of Clinical EFT on wellbeing, emotions, fear of cancer recurrence, and perceptions of recurrence in cutaneous melanoma survivors in remission, comparing changes over time between EFT and a control condition.To assess whether EFT instruction in a group setting, which has the potential to provide social support, makes this efficient mode of implementation non-inferior or even beneficial in comparison to personal instruction.To assess changes in SUD scores during EFT sessions.

## 2. Materials and Methods

### 2.1. Study Population

Study participants were adults who had previously been diagnosed with cutaneous melanoma by a private dermatology clinic that specialized in the early detection of skin cancer. Melanoma staging for study participants was conducted based on Breslow tumor thickness, a standard measure of vertical tumor depth. Among the participants, 42.3% were diagnosed with melanoma in situ, 34.6% had thin melanomas (0.1–0.7 mm), and 23.0% had melanomas of intermediate thickness (0.8–1.2 mm). No participants presented with regional lymph node involvement or distant metastasis and none of the participants had received additional oncological treatments such as immunotherapy, targeted therapy, chemotherapy, or radiation. In the present study, we refer to the participants as melanoma survivors, i.e., as individuals who had completed active surgical treatment for melanoma and were in the follow-up phase of care. Participants could also have been treated for concomitant additional non-melanoma skin cancers including BCC and Squamous Cell Carcinoma (SCC). Inclusion criteria were being aged over 18, and receiving a melanoma diagnosis more than six months prior to the study, as confirmed by clinical dermoscopic examination, and pathology results. Exclusion criteria were being in active treatment for melanoma; suffering from a psychiatric disease or epilepsy; or having been diagnosed with other types of malignant disease such as non-skin cancers.

### 2.2. Sample

The sample included 53 participants randomized between three study conditions: (1) Group EFT instruction and practice (G-EFT; *n* = 16); (2) Individual EFT instruction and practice (I-EFT; *n* = 18); and (3) waiting-list control condition (CC; *n* = 19). Participants’ mean age was 61.9 (±12), and 71.7% were females. Their number of years of education ranged from 12 to 23, and 74% held an academic degree. About one third were retired, while the rest were working part- or full-time. Most of the participants (88%) were married or cohabiting, and 98% were parents. As for level of religiosity, most participants (89%) were secular. Participants had been diagnosed with melanoma one or more years earlier, with a mean of 8.81 (±5.49). The demographic characteristics of the sample by study condition are presented in Table 1.

### 2.3. Recruitment and Procedure

This study was approved by the Tel Aviv University Institutional Review Board (approval #0004611) and registered at clinicaltrials.gov (Identifier: NCT05421988) (https://clinicaltrials.gov/study/NCT05421988?term=NCT05421988&rank=1). Potential participants meeting the inclusion/exclusion criteria received an email inviting them to participate in an online meeting presenting the study and its goals as well as information about the use of EFT as a practice for self-care and empowerment. Interested people were asked to respond by email. A week later, potential participants who had not responded were contacted a second time.

The online lecture covered the role of stress and emotions in disease and health and the basics of EFT. Participants were introduced to the theory and practice of EFT as described in The EFT Manual [26] and the structure and protocol of the study were explained in more detail. Following this lecture, participants interested in joining the study were asked to sign the informed consent form and those who consented proceeded to the randomization process.

Randomization to the study conditions was conducted in one step after consent and before the collection of baseline data. All fifty-six patients who consented to participate in the study were assigned code numbers and these were used in a computerized macro in Excel to randomize participants to the study conditions in a single step. Once randomized, the code numbers were converted back to the names to be invited to the sessions in each condition. Participants were assigned to one of the three study conditions, so that at least 18 participants would be included in each condition. The final sample for the G-EFT condition comprised only 16 participants of the 19 allotted to this condition, as after randomization one reported being newly diagnosed with a different form of cancer, one dropped out because his wife gave birth, and one ceased participation at the beginning of the study. Participant flow through the stages of the study is presented in the CONSORT diagram in Figure 1.

In both intervention conditions, EFT instruction included four weekly face-to-face sessions. G-EFT sessions took place in small groups of 7–9 participants each. Thus, neither the participants in the intervention conditions nor the EFT instructor could be blinded to their condition. CC participants were informed that they would be invited to attend group sessions in a few months yet would be asked to fill in questionnaires at several time points until these sessions begin. Thus, they were unaware of their allocation to a (waiting-list) control group. All study participants were asked to fill in questionnaires at three time points: (T1) at baseline (before the instruction sessions); (T2) at the end of the four-week instruction period; and (T3) three months later. The participants completed the questionnaires with the baseline and outcome measures through Qualtrics, a secure online survey platform. These questionnaires were filled out in private, at a time and place of the participants’ convenience, so no external assessor was involved. Subjective units of distress ratings were collected by the EFT instructor at the beginning and end of each session (i.e., these ratings were not filled in confidentially and the instructor was not blinded to the study condition).

The EFT instruction sessions were carried out between December 2022 and June 2023. The T3 questionnaires were filled in between April and August 2023. In each month of the study period, either Individual EFT (I-EFT) sessions or the Group EFT (G-EFT) intervention condition were carried out, alternating between them each month. In parallel, an equal number of control group participants were invited each month to complete the same set of questionnaires at the same times as the intervention groups. By the end of the post-intervention 3-month follow-up period, the questionnaires from both intervention and control participants had been completed. Then the CC participants received their instructional EFT sessions.

### 2.4. EFT Instruction and Practice

The EFT practitioner was trained and certified in Clinical EFT by EFT Universe (EFTuniverse.com). This one-year certification program trains participants in Clinical EFT as defined using the quality standards of the APA Division 12 Task Force for Empirically Validated Therapies [19]. EFT was applied with fidelity to the manualized form of the method as described in The EFT Manual [26].

In the G-EFT and I-EFT, the EFT practitioner met with the participants once a week for a 90 min session for 4 consecutive weeks. Participants were instructed to practice EFT daily using a predetermined Setup Statement and Reminder Phrase for 4 weeks. The G-EFT and I-EFT sessions were conducted face-to-face, except for one G-EFT session conducted online because of the possibility of COVID-19 exposure in a participant.

In the first session the practitioner facilitated participants’ listing of traumatic life events during childhood or adulthood which might have been related to the development of melanoma. The emotions and their intensity on an SUD scale were recorded in all sessions.

The second session was dedicated to performing EFT on one particular emotional event chosen by each participant using Setup Statements that were highly specific for that participant. In the third session, EFT practice focused on the emotions associated with fears of melanoma, alternating between particular fears and general fears. The fourth session was dedicated to applying EFT to emotions related to disease perception and the healing process.

In the G-EFT, the first session was also individualized by being dedicated to examining traumatic life events and the emotions they elicited. The benefits of performing EFT in a group session were explained. In the next sessions, the EFT practitioner worked with one participant at a time while other participants practiced EFT, using either Setup Statements provided by the practitioner or generating their own. The themes of the sessions were the same as in the I-EFT, where EFT was performed alternating between the emotions and fears of specific participants and emotions and fears general to the group.

### 2.5. EFT Protocol

Participants typically began each session by identifying the traumatic life events they wished to address with EFT. After filling in a chart with the titles of triggering events and their related emotions, they chose the first one to tap on. They rated their emotional arousal on a scale of 0 (minimum distress or neutral state) to 10 (the maximum amount of distress).

Participants then incorporated the title of their event into a “Setup Statement”, which focused their attention on their distress (the exposure component of EFT). The Setup Statement was typically stated in the following format: “Even though I have this problem (e.g., my brother hit my face with a rock when I was 4), I deeply and completely accept myself.” The first half of the Setup Statement emphasizes exposure, whereas the second half frames the traumatizing event in the context of self-acceptance. An acupoint on the outside of the palm of the hand is tapped while the Setup Statement is repeated three times. The “Reminder Phrase” produces continued exposure and emotional arousal while the participant taps on a sequence of face and body acupoints. Successive rounds of EFT are performed till the SUD rating drops to a zero or close to that number.

### 2.6. Instruments

Sociodemographic data included year and place of birth, gender, educational level, employment and financial status, and family status.

The health-related part of the survey included sections on general health, comparison with peers of the same age, smoking habits, alcohol consumption, and physical activity. Additional inquiries covered chronic medical conditions in detail. The specifics for each section are outlined below.

Health status was assessed with two commonly used self-ratings [77]. Participants rated their overall health from 1 (poor) to 5 (excellent), and their health in comparison to peers of their same age from 1 (much worse) to 5 (much better).

Health behaviors included the following:

Skin cancer risk was assessed with a single item asking about the number of hours on average they spent outdoors in the past week during daytime, and seven additional items from the risk scale of the Sun Protection Questionnaire [78]. These items included questions about skin reactions to sun exposure, hair and eye color, freckles, moles, and family history of skin cancer. Responses were assigned values ranging from 0 to 2, with a higher score indicating an elevated risk level [78]. The coded responses were summed to create a skin cancer risk score.

Smoking was categorized as having never smoked, being a past smoker, or being a current smoker.

Alcohol consumption included a question about whether they drink any alcohol. A positive answer was followed by questions about the number of drinks consumed during the average week from the categories of beer, wine, and other alcoholic beverages. The responses were summed.

Physical activity questions referring to the past three months were adapted from Godin and Shephard [79]. Participants were asked about the following: (1) the frequency of their engagement in exercise each week; (2) whether it fell into the light, moderate, or intense category, and (3) the duration in minutes of each session. For each level of activity, we multiplied the number of sessions per week by the duration of each session. These scores were multiplied by 3, 5, and 9, for light, moderate, and intense activity in order to measure Metabolic Energy Expenditure Units (METs) [79]. The three scores were then summed to create a total estimate of the person’s engagement in physical activity, which factored in both the duration and the intensity of activity.

Wellbeing was assessed using the multidimensional approach of Bonnachi et al. [51]. This approach includes ratings of physical, psychological, spiritual, relational, and general wellbeing. Participants were asked to rate each of these dimensions on a numerical scale from 1 (absolute distress) to 10 (complete wellbeing). We also computed a mean of the five ratings (α = 0.76).

Affect measures included a single item assessing happiness from 0 (deeply unhappy) to 10 (very happy) [80] and a short form of the brief positive and negative affect scales (PANAS; [81]), translated to Hebrew [82]. This measure included five positive feelings (e.g., excited, inspired) and five negative ones (e.g., nervous, distressed), rated for their frequency in the past two weeks on a scale from 1 (not at all or very little) to 5 (very much). Both scales showed good internal reliability (α = 0.85 and α = 0.75 for positive and negative affect, respectively). Mean scores were computed for positive and for negative affect.

Fear of cancer recurrence was assessed with the Fear of Cancer Recurrence Inventory—Short Form (FCRI-SF), with permission from the authors [83,84] and the translators of the Hebrew version [85,86]. This measure includes nine items: Four statements about concerns regarding cancer recurrence and one about the belief that the cancer will not recur (reverse-coded). Participants are asked to rate how much each of these five statements describes their feelings in the past month on a scale from 0 (not at all) to 4 (very much). Four additional items ask about self-perception of risk of recurrence, frequency of thoughts about the possibility of recurrence, how much time each day is spent thinking about the possibility of recurrence, and the length of time for which such thoughts have occupied a participant’s mind. Each item is rated on a 0–4 scale. The nine-item scale showed high internal reliability (α = 0.84). Mean scores were computed.

Recurrence illness perceptions were assessed based on the Revised Illness Perception Questionnaire (IPQ-R), which was adapted and validated with cancer patients [87]. It was later adapted specifically for cancer survivors, in reference to perceptions of recurrence [88]. This instrument enables the measurement of perceptions of cancer and its recurrence before and after psychological interventions. We further adapted the questionnaire to focus specifically on perceptions of the recurrence of melanoma.

The questionnaire includes four subscales: consequences—three statements about the limitations imposed on work, sports, or family activities because of the need to limit sun exposure (α = 0.84); personal control—seven statements about the steps a participant can take that affect their risk of cancer recurrence (two items were reverse-coded; α = 0.69); coherence—four statements measuring the participant’s degree of understanding of how to minimize the risk of cancer recurrence (two items were reverse-coded; α = 0.69); and emotional representations—four statements about the participant’s emotional reactions to the possibility of cancer recurrence (α = 0.90). All statements were rated on a 5-point Likert scale from 1 (strongly disagree) to 5 (strongly agree). Mean scores were computed for each of the four subscales.

General self-esteem was assessed with the Chen et al. [89] General Self-efficacy Scale. This measure includes eight statements about one’s confidence in their ability, rated on a 5-point Likert scale from 1 (strongly disagree) to 5 (strongly agree). The scale showed high internal reliability (α = 0.90). Mean scores were computed.

Acute distress was measured using Standard Units of Distress (SUDs), an 11-point scale ranging from 0 (minimal distress) to 10 (maximum distress), which has long been applied in many therapeutic settings as a subjective measure assessing a participant’s discomfort and emotional and somatic markers such as anxiety, fear, pain, dysfunctional beliefs, and emotional memories [90]. Increased SUDs are associated with heightened arousal of the sympathetic nervous system [91], while SUDs also correlate with respiratory rate, heart rate, and galvanic skin response [92]. When psychological treatments are successful at lowering reported SUDs, physiological markers of stress are also reversed [93].

### 2.7. EFT Practice and Satisfaction

Participants in both EFT instruction conditions were asked in the T2 and T3 (online) questionnaires how many times they had practiced the EFT technique in the past 7 days, followed by an open-ended question about their impressions of using the technique.

In the T3 questionnaire, participants from the EFT instruction conditions were asked to rate their satisfaction level with the EFT technique training from 1 (not at all) to 5 (very much). Additionally, they were asked whether they would recommend learning and practicing EFT to another patient with a similar diagnosis (yes/no). Two open-ended questions allowed participants to further elaborate on their subjective experience with EFT instruction and practice, including aspects they appreciated and any changes they would suggest.

### 2.8. General Notes

Questionnaires that were not available in Hebrew were translated and back-translated by researchers fluent in both languages. Demographics, health, and health behavior questions were only asked at baseline. Questions about the practice of EFT and satisfaction with training were only asked at the follow-ups and only in the two intervention conditions. All other questionnaires were included at all three time points.

### 2.9. Statistical Analyses

The sample size was determined a priori using G*Power 3.1.9.7 (Franz Faul, Kiel University, Germany) [94]. For repeated measures ANOVA with a within–between interaction, three study conditions, and three measurements, a sample of 45 (15 per condition) was found to provide a power of 0.90 to detect a medium-sized effect (d = 0.50) at a significance level of alpha = 0.05 or lower. Projecting a 20% drop-out rate, we aimed to recruit 57 participants (19 per condition) and to first conduct analyses per protocol with all available data for each analysis. If drop-out rates were substantial, we planned to also conduct Intention-to-Treat (ITT) analyses with the full sample (using the last available data point carried forward). As described above, we recruited and randomized 56 participants, and the baseline sample included 53 participants. Since drop-out rates over time were very low (one person in the CC did not fill in the questionnaire at T2 and one person in the G-EFT condition did not fill in the questionnaire at T3), we only conducted analyses per protocol: Cross-sectional comparisons between the study conditions and the effect sizes for comparisons among the study conditions in terms of changes from beginning to end of study were conducted with data from 52 participants, and comparisons across all three time points were carried out on the final analysis sample of 51 participants. The actual power for detecting a medium-sized effect was 0.95.

Analyses were conducted with IBM SPSS Statistics for Windows, version 29 [IBM Corp, Armonk, NY, USA]. Among the participants who filled in each questionnaire, there was no construct-level missingness. We first compared the baseline characteristics of the participants in the three conditions using one-way ANOVAs, with post hoc tests employing a Bonferroni correction for multiple comparisons. Changes over time were tested with two-way repeated measures ANOVAs with one between-subjects (study condition) and one within-subjects (time) factor. These planned analyses were repeated with one covariate (skin cancer risk factor score, which was found to differ among the conditions at baseline, despite randomization; see below) and these analyses are presented in the Results section. Significant time by condition linear and/or quadratic interactions indicated a significantly different pattern of changes across the three time points among the three conditions. We considered Mauchly’s test of sphericity, for which a significant result would mean that the assumptions on which this mixed-model approach was based were violated. The test yielded non-significant results for all but two measures: happiness and spiritual wellbeing. In both cases, the Greenhouse–Geisser epsilon value was >0.80. For these two measures, sphericity was not assumed and we used the Huynh–Feldt adjustment (which is recommended when the Greenhouse–Geisser epsilon value is > 0.75; in practice we also computed the findings with the Greenhouse–Geisser adjustment and the results were very similar). Since the hypotheses were directional, expecting improvements in the experimental conditions in comparison to the control, the significance of the differences among the three conditions in the trends over time was defined as *p* < 0.05 in one-tailed tests, and the 90% confidence intervals (CI) are shown in the figures.

## 3. Results

### 3.1. Health and Heath Behaviors at Baseline

The participants reported overall good health, with a mean of 4.00 (±0.56) out of a possible 5. For health compared to age peers, the mean rating was 3.51 (±0.72). There were only small, non-significant differences in these ratings among the three conditions. Sixty-four percent of the sample (*n* = 34) had never smoked, five participants currently smoked (one in the G-EFT, four in the CC), and fourteen were past smokers (three, six, and five, in the G-EFT, I-EFT, and CC, respectively). About half the sample (49%) reported that they did not drink any alcohol; the remainder were mostly light drinkers: only three participants in the G-EFT, four in the I-EFT, and three in the CC reported drinking more than two drinks a week on average. As for physical activity, 43 participants (81%) reported engaging in physical activity in the past three months. The overall level of activity did not differ among the three conditions. Six participants in the G-EFT, one in the I-EFT, and three in the CC reported not engaging in physical activity in the past three months.

The conditions differed in their skin cancer risk factor score. The mean (±SD) score was 8.25 (±1.91) for G-EFT, 5.83 (±2.54) for I-EFT, and 7.37 (±2.85) for CC (F (2, 50) = 4.12, *p* = 0.02). Post hoc tests with a Bonferroni correction showed that the skin cancer risk score in the G-EFT was significantly higher than in the I-EFT. Therefore, all longitudinal analyses were conducted controlling for the skin cancer risk scores.

Descriptive statistics of the study measures at baseline appear in Table 2. The three conditions did not differ in any of the study variables at baseline (T1), with one exception: emotional representations. Post hoc comparisons with a Bonferroni correction showed that the I-EFT participants reported significantly lower (less negative) emotional representations of cancer recurrence compared to the CC (*p* = 0.003)

### 3.2. Differences Among the Study Conditions over Time

Changes over time were tested with two-way repeated measures ANOVAs with one between-subjects factor (study condition) and one within-subjects factor (time), adjusted for skin cancer risk scores. A significant time by condition interaction indicated significantly different patterns of within-subject changes along the three points in time in the three conditions. In perceptions of cancer recurrence, such an interaction was found for coherence and for emotional representations.

Coherence (understanding how one can prevent recurrence) showed a significant improvement over time (F (2, 94) = 3.33, *p* = 0.04 one-tailed), which was qualified by a significant time by condition interaction (F (4, 94) = 4.14, *p* = 0.004 one-tailed). This interaction reflected an improvement in coherence in the two EFT conditions, particularly in the G-EFT, from before (T1) to after the intervention period (T2), in contrast with a slight decline during the equivalent time in the CC. This was followed by stability between T2 and T3 in all three conditions. The interaction consisted of both linear (F (2, 47) = 4.77, *p* = 0.01 one-tailed) and quadratic (F (2, 47) = 3.39, *p* = 0.04 one-tailed) effects. These patterns can be seen in Figure 2.

Emotional representations also showed a significant change over time (F (2, 94) = 4.07, *p* = 0.02 one-tailed), which was qualified by a significant time by condition quadratic interaction (F (2, 94) = 3.82, *p* = 0.03 one-tailed). As can be seen in Figure 3, this interaction reflected a more complicated pattern: the CC showed a decline to T2 in negative emotional representations of recurrence and then a slight increase, while the other two conditions showed a relatively stable pattern over time, with only very small changes. Further comparisons showed an overall significant difference in the pattern of changes only between the I-EFT and the CC, and an interaction revealed there was only a different quadratic (F (2, 47) = 3.82, *p* = 0.01 one-tailed) and not a linear pattern of change over time in these conditions.

Fear of cancer recurrence showed a slight, statistically non-significant, and linear decrease over time in all three conditions. Personal control and consequences showed small and non-significant changes over time.

Among the affect and wellbeing measures, a time by study condition interaction was found for spiritual wellbeing and for happiness. For all three conditions, positive and negative emotions, the other four dimensions of wellbeing, and general self-esteem all showed only small, non-significant changes over time. Spiritual wellbeing improved in both EFT conditions from T1 to T2, then stabilized (with a slight further increase in the I-EFT). In contrast, the CC remained at about the same level across the same time period (see Figure 4). This pattern of change reflected a significant overall time effect (F (2, 94) = 4.73, *p* = 0.02 one-tailed), which was qualified by a significant linear time by condition interaction (F (2, 47) = 4.25, *p* = 0.02 one-tailed) but not a quadratic pattern.

For happiness, a more complicated pattern appeared (see Figure 5): The G-EFT showed a decline from T1 to T2, followed by a steeper increase to T3, while both the I-EFT and the CC showed small changes over time. The overall time effect was not significant, yet the time by condition interaction reflected a significant difference among the conditions in the pattern of changes over time due to a significant quadratic (F (2, 47) = 3.75, *p* = 0.03 one-tailed) but not linear difference in the patterns of change across time.

To assess trends over time across all study measures, we computed effect sizes (Hedge’s g, which corrects for samples smaller than 20) for the changes in each variable from baseline (T1) to the end of follow-up (T3), three months after the end of the intervention. We computed these effect sizes separately for each of the experimental conditions and at an equivalent time for the CC. These computations were conducted once for the full sample and a second time in the experimental conditions but only for those who practiced EFT at T3 at least once a week. The scientific conventions set by [95] define 0.20 as a small effect size and 0.50 as a medium effect size. These effect sizes are presented in Table A1 in Appendix A. Many of these changes were not statistically significant in the full sample, as reported above, and with the small sample sizes for those who practiced they were even less likely to be significant. Nevertheless, these findings may contribute to a deeper understanding of the possible effects of the EFT intervention, as can be seen in several trends that are worth noting: First, the effect sizes for the CC were mostly very small. Second, the effect sizes were generally stronger for the G-EFT than for the I-EFT. Third, the effect sizes were stronger for some of the variables among those who were still practicing EFT at T3.

### 3.3. Changes in Subjective Unit of Distress (SUD) Scores

We compared participants’ SUD scores at the beginning of each session to their ratings at the end of the session. The differences were dramatic, showing that all sessions achieved the goal of reducing distress related to the issue chosen for EFT practice during the session. SUDs at the beginning of a tapping session averaged 7–8 on the 10-point scale and only 0–1 at the end. These marked decreases in SUDs were observed in both intervention conditions at all four sessions: The average decrease was 6.8 points, a difference that was highly statistically significant (*p* < 0.0001). On average, this amounted to a 90% decrease compared to the SUD at the beginning of a session.

### 3.4. Participant Satisfaction with EFT Instruction and Practice

The two intervention conditions did not differ in their ratings of satisfaction or in the extent to which they continued to practice EFT. Satisfaction with EFT instruction and practice was very high in both conditions, as can be seen in Table 3: 20 of the 33 participants reported being extremely satisfied, 11 reported high satisfaction, and only 2 (in the I-EFT condition) reported moderate or lower satisfaction. In addition, 32 of 33 participants reported at T3 that they would recommend EFT to a patient in a situation similar to theirs.

Despite these very high levels of satisfaction, not all participants continued to practice EFT (see Table 4). At T2, shortly after the intervention ended, 7 of 34 participants did not practice at all and only 8 practiced almost daily. At T3, three months later, more than a third of the participants were not practicing, and the remainder practiced 1–5 times a week. None were practicing daily.

Participants’ responses to the open-ended question about their impressions of EFT were mostly very positive, regardless of the frequency of practice. For example, they reported that this technique lowered stress and worry, made them aware of the cause of their stress, provided calmness and confidence, improved their emotions and thoughts, helped them cope with sleeping difficulties, and increased energy and feelings of control. At T2, 13 of 15 participants in the G-EFT and 16 of 18 in the I-EFT affirmed positive changes and satisfaction. At T3, many participants were no longer practicing EFT yet all except one participant in each condition reported positive impressions. Those who were no longer practicing mentioned that it was helpful when they had practiced.

When asked at T3 about their experience of the EFT instruction and practice, 14 participants in each intervention condition responded. All but one described a very positive experience, including those who were rarely or not at all practicing EFT (some of whom mentioned that they would like to use it more). Several participants in the G-EFT mentioned that the 4-week instruction period was too short and it would have been helpful to have continued to meet occasionally. In fact, one of the groups in this condition collectively decided that they would continue to meet on their own.

## 4. Discussion

This is the first study to examine the effects of EFT training and practice in melanoma survivors. EFT instruction was very well received by participants, according to their open- and closed-ended reports of satisfaction. The open-ended responses showed that 81% of the G-EFT and 89% of the I-EFT reported positive changes and satisfaction at T2. At T3, 97% of the participants in the two intervention conditions stated that they would recommend EFT to a person in a situation similar to theirs. The emotional distress surrounding traumatic events that participants considered to be related to their melanoma was also ameliorated, as exhibited by the reduced SUD scores of almost all participants during each EFT session.

However, these very high levels of satisfaction and the decrease in distress during the sessions were accompanied by changes in only a few of the emotional and cognitive outcomes measured. Most of the measures we investigated remained stable across the intervention period and the following three months and did not show statistically significant improvements in comparison to the CC.

Adverse childhood experiences (ACEs) have been linked to the development of chronic diseases in adulthood and certain types of cancer [96,97]. A number of studies have focused on exploring the potential influence of stress on the manifestation and advancement of melanoma [16,18,98,99]. However, research on the specific link of ACEs to skin cancer is limited [13,14], and there is a lack of detailed investigation specifically examining the interconnection between ACEs, adult traumas, and the development of melanoma.

In our sample of melanoma participants, the assessment of possible traumatic events during both childhood and adulthood was conducted during the first EFT meeting. Following the manualized form of the method [26], instruction was tailored to address the specific traumatic events reported by each participant. Many participants perceived a causal connection between particular traumas and the etiology of their melanoma. They reported this link as a motivating factor in enrolling for this study.

Two important results were observed: There was a significant increase in coherence in both EFT intervention conditions, particularly in the G-EFT, compared to CC. Both intervention conditions maintained their higher level of coherence over the next three months. In the current context, the term “coherence” refers to the extent to which a participant understands the measures that need to be taken to prevent cancer recurrence. Most of the participants were diagnosed years ago and were aware of the need to limit sun exposure yet were not aware of the possible effects of stress on disease progression. The EFT sessions may have provided an opportunity to address recurrence and its prevention in a wider perspective that included emotional and spiritual wellbeing, along with other well-known factors related to the etiology of melanoma.

The second significant change observed was the increase in spiritual wellbeing in the two intervention conditions during the 4-week intervention period. The finding that improved spiritual wellbeing remained high in the following three months, and even further increased in the I-EFT, was of special interest. EFT may facilitate the ability of participants to address their spiritual needs. Other studies of spiritual needs [100,101], as well as a Danish study that was the largest ever [100], identified associations between spiritual needs and self-rated health and wellbeing. Additionally, studies have found that addressing spiritual needs can improve patients’ health outcomes [102,103].

In this era of integrative medicine and a holistic view of health that extends beyond biological and psychosocial wellbeing, there is a growing focus on the spiritual dimension of wellbeing. Numerous studies have demonstrated the positive impact of spirituality on physical and mental health. This includes dimensions such as subjective wellbeing, health-related quality of life, coping skills, mental illness, addictions, professional productivity, and suicidal behaviors [53,104,105,106,107,108].

Research has also shown that spiritual engagement can positively affect adjustment to cancer [109,110,111]. Furthermore, a previous study demonstrated that 77% of patients with cancer wanted spirituality to be part of their caregivers’ routine focus [112]. Taking into account that spiritual wellbeing is a significant unique contributor to QOL, beyond the core domains of physical, social/family, and emotional wellbeing [51,113], our finding of improved spiritual wellbeing points to a possible improvement of the overall QOL of the participants in the long term. This is consistent with recent findings in the literature [53].

Negative emotional representations of recurrence remained stable in the two intervention conditions. The CC participants reported lower levels in the second compared to the first time point (one month apart) and then an increase across the next three months, though they did not return to the baseline level. The direct measure of fear of recurrence also remained quite stable across time in the two intervention conditions. These findings reflect those of a large multinational study of breast cancer patients in which fear of recurrence was also found to be consistent in the first 18 months post-diagnosis [85].

Happiness remained consistent in the I-EFT and CC, whereas the G-EFT reported a short-term decrease in happiness by the end of the intervention and then an increase. This increase may reflect social ties between some of the group participants, who elected to continue their meetings independently after the intervention period ended.

Although participants reported very high levels of satisfaction, many of them did not continue to practice EFT daily. The majority did continue at least once or twice a week. Some of those who did not continue stated that they would like to. An earlier study of 216 healthcare workers observed a similar pattern of sharp symptom declines after EFT but that subsequent regular tapping was only conducted by a minority of participants [32]. Two thirds of the participants in the current study were employed and may have found it difficult to integrate a new activity into their lives on a regular basis without the support of an instructor and community. Family and routine activities may also have made instilling a new habit challenging.

Repeating the analyses without the participants who did not continue practicing EFT after the end of the intervention resulted in an even smaller sample size, so that any changes identified were not significant. However, the analyses of effect sizes of changes along the study showed several trends which were not statistically significant yet were consistent across many variables and hence worth noting for future studies. Group EFT seemed to be more effective than individual instruction of EFT, and both tended to be more effective in comparison to the control condition. Continued EFT practice tended to show better results. Therefore, future studies should investigate how to optimize the effects of EFT training in a group setting and whether ‘booster’ group meetings could contribute to continued EFT practice and potentially to more positive outcomes and their maintenance over time.

Even though EFT is a novel therapy in oncology, numerous studies acknowledge this technique as a well-established evidence-based practice for both physiological and psychological symptoms in various populations [20,23,38,114]. Although there is clinical research supporting the effectiveness and benefits of EFT in the realm of emotional and psychological wellbeing and mental health, as well as in physical pain and improvement in markers of physiological health, its role in enhancing the wellbeing of cancer patients is still in the nascent stage of investigation. The existing studies show positive effects on mood, stress, and wellbeing [46,48,49,50,115], though none of these examine a population of melanoma patients.

Our study had several limitations. First, our sample size was small. It was powered to detect medium or larger effect sizes, yet most measures showed smaller effects. Spiritual wellbeing and coherence showed large effect sizes, like previous studies on the effects of EFT on mental health and in correspondence with the salience of these two topics in the intervention sessions. The modest effect sizes observed for other measures may have also been due to many participants failing to continue practicing EFT regularly.

Second, the current sample was drawn from a demographically and psychologically resourceful population with high general self-efficacy, good perceived health, and beneficial lifestyle behaviors. And, for ethical, scientific, and practical reasons, the participants were mostly diagnosed years ago, and had been under the regular care of a physician with awareness of resilience issues. These two characteristics of the study population suggest that our findings underestimate the full potential of EFT for the general population of melanoma cancer survivors. Future studies should include patients more recently diagnosed with melanoma and from more diverse populations.

Third, outcomes were assessed only with self-reported measures, because we aimed for changes in wellbeing, a subjective evaluation, yet such measures may be prone to various biases. SUD scores may also be biased by being collected in the session by the instructor. However, all longitudinal outcome measures were collected confidentially online. Fourth, there was no way of blinding the study conditions to the participants or the practitioner, which could have introduced biases due to their expectations. Moreover, the CC participants also attended the initial lecture about stress and health and the research showing that EFT has a positive effect on health, then completed the questionnaires at baseline and over the following months while waiting to take part in the intervention. This may have raised their awareness of recurrence issues and led them to address their wellbeing in other ways, which might have diminished the differences between them and the intervention conditions. Finally, from January 2023 and throughout the study period, Israel underwent a period of social and political unrest. In addition, in May 2023, rocket attacks affected many cities in Israel. This context, with internal tensions and external pressures, may have affected all participants’ wellbeing and diminished group differences and the effects of EFT.

This study indicated the potential contribution of EFT to the wellbeing of melanoma patients. It sensitized participants to the impact of traumatic events, whether they occurred in childhood or adulthood, and highlighted the need to address trauma as a crucial part of a comprehensive integrative melanoma treatment and prevention plan. The marked reduction in immediate emotional distress as evidenced by in drops in SUD scores and the participants’ satisfaction with the EFT method showed that EFT can be an effective stress relief and coping technique for melanoma patients.

This study also suggests that EFT could be a valuable tool for improving the overall quality of life for melanoma survivors by enhancing their spiritual wellbeing. The improved understanding of the prevention of cancer recurrence could contribute over time to the greater adherence to preventive measures and consequently to less fear of recurrence. Future studies could investigate the effects of a longer period of instruction and practice, and to greater support for continuing to tap long-term, which may show more lasting effects on the reduction in negative emotions and the promotion of positive ones.

This study was one of the first to make a direct comparison between group instruction in EFT and individual therapy. It showed that group application was just as effective and satisfying as individual instruction. This suggests that the diminution of individual attention in a group setting may be offset by the benefits of peer support and group dynamics. Group instruction is more efficient in terms of instructor time and effort and may allow for the expansion of EFT training to larger numbers of patients.

## 5. Conclusions

Our study offers evidence that teaching EFT to melanoma patients as a non-pharmacological and non-invasive self-help method is well-received and may be effective in reducing psychological stress, raising patient awareness of possible links between traumatic emotional events and wellbeing, and improving spiritual wellbeing and the understanding of measures that reduce the risk of cancer recurrence. The findings point to the need for further research on ways to optimize EFT training for this population and encourage adherence over time to increase its potential benefits.

## Figures and Tables

**Figure 1 healthcare-13-01420-f001:**
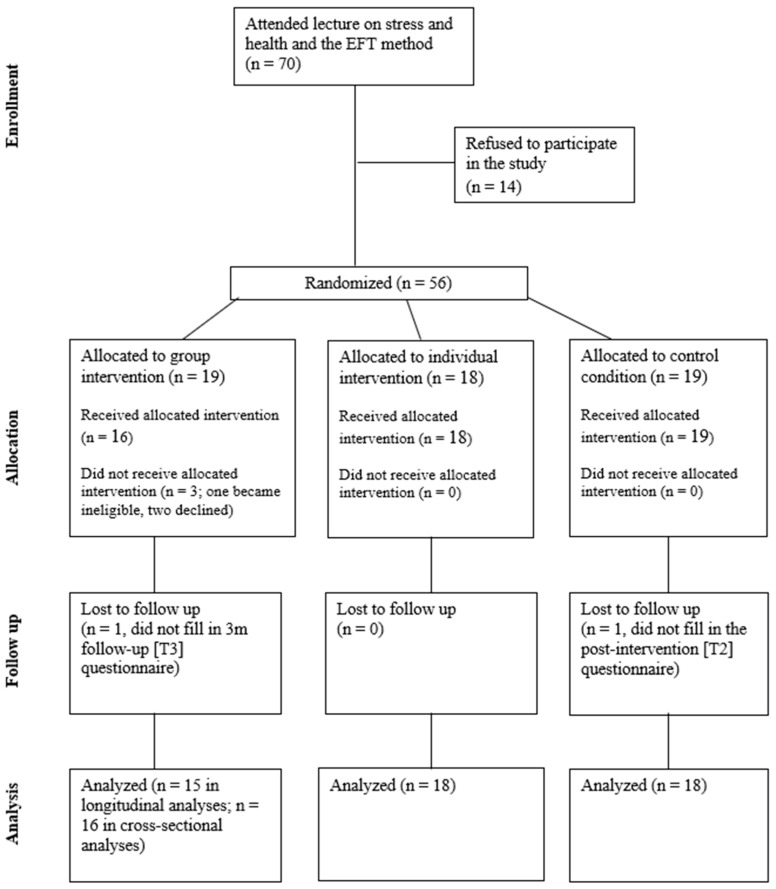
CONSORT diagram showing the flow of participants through each stage of the randomized trial.

**Figure 2 healthcare-13-01420-f002:**
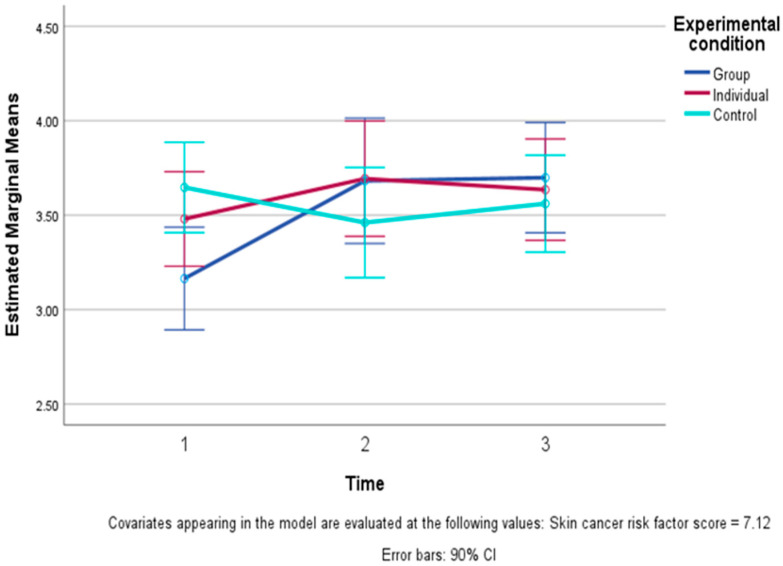
Levels of coherence (understanding prevention of recurrence) and 90% confidence intervals (CIs) across time in the three conditions: Group EFT training, Individual EFT training, and control condition (controlling for skin cancer risk scores). The figure presents a significant linear and quadratic study condition X time interaction, i.e., a significant difference (*p* < 0.05) between the pattern of improvement and then stability in the coherence levels in the two EFT conditions and the more stable scores over time in the control condition.

**Figure 3 healthcare-13-01420-f003:**
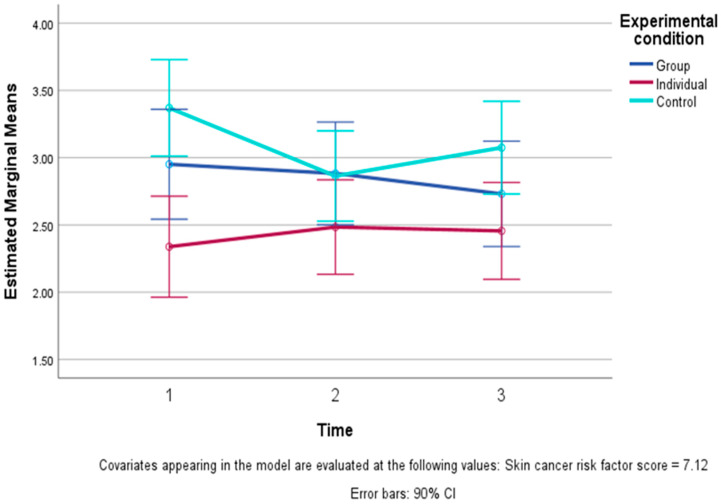
Levels of emotional representations (negative emotional reactions to the possibility of cancer recurrence) and 90% confidence intervals (CIs) across time in the three conditions: Group EFT training, Individual EFT training, and control condition (controlling for skin cancer risk scores). The figure presents a significant quadratic study condition X time interaction, i.e., a significant difference (*p* < 0.05) in the patterns of change in the three conditions, i.e., relative stability in the two EFT conditions compared to a decrease and then increase in emotional representation in the control condition.

**Figure 4 healthcare-13-01420-f004:**
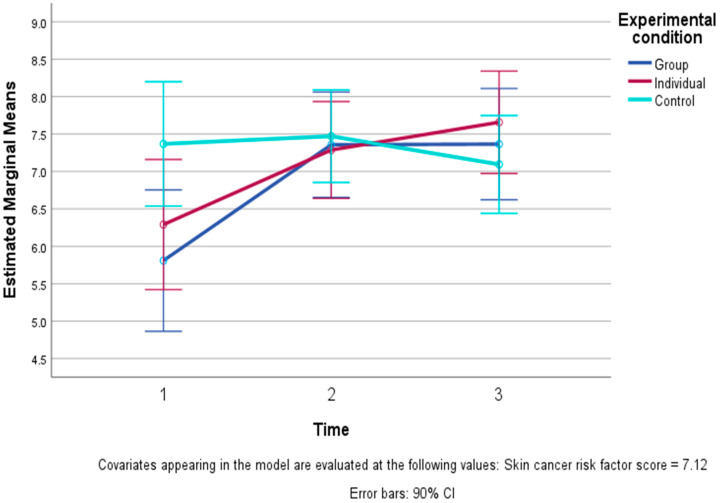
Levels of spiritual wellbeing and 90% confidence intervals (CIs) across time in the three conditions: Group EFT training, Individual EFT training, and control condition (controlling for skin cancer risk scores). The figure presents a significant linear study condition X time interaction, i.e., a significant difference (*p* < 0.05) between the pattern of steep improvement and then more stable spiritual wellbeing in the two EFT conditions compared to stability and then some decline in the control condition.

**Figure 5 healthcare-13-01420-f005:**
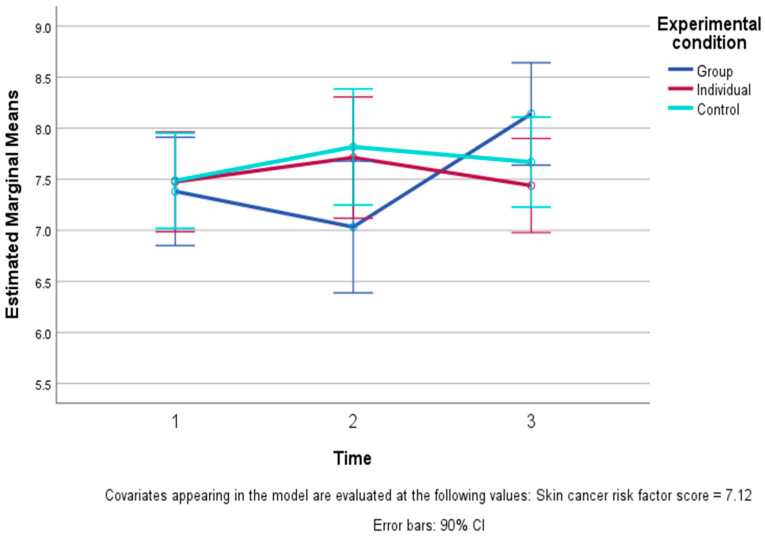
Levels of happiness across time and 90% confidence intervals (CIs) in the three conditions: Group EFT training, Individual EFT training, and control condition (controlling for skin cancer risk scores). The figure presents a significant quadratic study condition X time interaction, i.e., a significant difference (*p* < 0.05) in the patterns of change in the three conditions, i.e., relative stability in the Individual EFT condition and in the control condition, compared to a decrease and then a steep increase in the level of happiness in the Group EFT condition.

**Table 1 healthcare-13-01420-t001:** Sociodemographic characteristics of the participants in the three conditions ^a b^.

Variable	Group EFT Instruction (*n* = 16)	Individual EFT Instruction (*n* = 18)	Waiting-List Control Condition (*n* = 19)	All (*N* = 53)
	% (*n*)	% (*n*)	% (*n*)	% (*n*)
Gender				
Female	81.3 (13)	66.7 (12)	68.4 (13)	71.7 (38)
Male	18.7 (3)	33.3 (6)	31.6 (6)	28.3 (15)
Employment				
Employed	62.5 (10)	66.7 (12)	68.4 (13)	66.0 (35)
Unemployed or fully retired	37.5 (6)	33.3 (6)	31.6 (6)	34.0 (18)
Education level				
Partial high school	0.0 (0)	5.6 (1)	0.0 (0)	1.9 (1)
High school graduate	0.0 (0)	11.1 (2)	15.8 (3)	9.4 (5)
Non-academic higher education	18.7 (3)	11.1 (2)	15.8 (3)	15.1 (8)
Bachelor’s degree	31.3 (5)	33.3 (6)	31.6 (6)	32.1 (17)
Master’s or doctoral degree	50.0 (8)	38.9 (7)	36.8 (7)	41.5 (22)
Marital status				
Married or cohabiting	81.2 (13)	100.0 (18)	84.2 (16)	88.7 (47)
Separated, divorced, or widowed	12.5 (2)	0.0 (0)	10.5 (2)	7.5 (4)
Single	6.3 (1)	0.0 (0)	5.3 (1)	3.8 (2)
Income level (*n* = 51)				
Below average	6.7 (1)	5.9 (1)	5.2 (1)	5.9 (3)
Average	33.3 (5)	17.6 (3)	47.4 (9)	33.3 (17)
Above average	60.0 (9)	76.5 (13)	47.4 (9)	60.8 (31)
Religion				
Secular	93.8 (15)	88.9 (16)	84.2 (16)	88.7 (47)
Traditional or religious	6.2 (1)	11.1 (3)	15.8 (3)	11.3 (6)

^a^ The total number of respondents is 53, unless otherwise stated. ^b^ There are no statistically significant differences among the conditions.

**Table 2 healthcare-13-01420-t002:** Means and standard deviations (SDs) of the study measures in the three study conditions at the three time points.

Measure (Scale)	Study Condition ^a^	T1—Baseline		T2—End of Intervention ^b^		T3—Three Months Later	
		Mean	SD	Mean	SD	Mean	SD
Happiness (0–10)	Group	7.44	0.96	7.19	1.94	8.13	0.74
	Individual	7.39	1.34	7.61	1.20	7.44	1.42
	Control	7.58	1.17	7.83	1.04	7.74	0.99
	Total	7.47	1.15	7.56	1.42	7.75	1.12
Negative affect (1–5)	Group	2.18	0.67	2.04	0.90	2.24	0.46
	Individual	2.22	1.02	2.02	0.74	2.18	0.76
	Control	2.07	0.75	1.99	0.52	1.97	0.54
	Total	2.15	0.82	2.02	0.71	2.12	0.61
Positive affect (1–5)	Group	3.24	0.84	3.29	0.67	3.41	0.77
	Individual	3.16	0.68	3.12	0.85	3.09	0.84
	Control	2.97	0.61	3.03	0.62	3.03	0.79
	Total	3.11	0.71	3.14	0.72	3.16	0.80
Self-efficacy (1–5)	Group	3.87	0.51	3.97	0.39	4.03	0.44
	Individual	3.89	0.65	3.98	0.48	3.86	0.58
	Control	3.77	0.56	3.84	0.41	3.88	0.50
	Total	3.84	0.57	3.92	0.43	3.92	0.51
Wellbeing—mean of five items (1–10)	Group	7.15	1.07	7.49	0.99	7.88	0.75
	Individual	7.46	1.14	7.86	1.02	7.92	1.21
	Control	7.75	1.04	7.71	1.04	7.66	0.82
	Total	7.47	1.09	7.69	1.01	7.82	0.95
Physical wellbeing (1–10)	Group	7.50	1.32	7.50	1.15	7.73	1.22
	Individual	7.78	1.31	7.89	1.18	7.94	1.39
	Control	7.63	1.46	7.72	1.18	7.47	1.12
	Total	7.64	1.35	7.71	1.16	7.71	1.24
Mental wellbeing (1–10)	Group	7.25	1.24	7.19	1.83	7.93	0.88
	Individual	7.39	1.91	7.78	1.31	7.61	1.54
	Control	7.84	1.39	7.44	1.25	7.89	0.88
	Total	7.51	1.54	7.48	1.46	7.81	1.14
Relational wellbeing (1–10)	Group	7.75	1.43	7.94	1.18	8.27	1.16
	Individual	8.28	1.02	8.33	0.77	8.50	0.79
	Control	8.26	1.28	8.17	1.25	8.00	1.45
	Total	8.11	1.25	8.15	1.07	8.25	1.17
Spiritual wellbeing (1–10)	Group	5.81	2.37	7.13	1.20	7.47	1.46
	Individual	6.17	2.15	7.44	1.62	7.56	1.79
	Control	7.11	2.08	7.44	1.79	7.16	1.61
	Total	6.40	2.22	7.35	1.55	7.38	1.61
General wellbeing (1–10)	Group	7.44	1.03	7.69	0.79	8.00	0.65
	Individual	7.67	1.09	7.83	0.99	8.00	1.41
	Control	7.89	1.05	7.78	1.06	7.79	0.79
	Total	7.68	1.05	7.77	0.94	7.92	1.01
Fear of cancer recurrence (0–4)	Group	1.57	0.65	1.43	0.56	1.42	0.69
	Individual	1.19	0.87	1.14	0.76	1.10	0.73
	Control	1.56	0.48	1.36	0.63	1.40	0.65
	Total	1.43	0.69	1.30	0.66	1.31	0.69
Consequences (1–5)	Group	3.18	1.44	2.60	1.33	2.80	1.40
	Individual	2.28	1.04	2.56	1.27	2.52	1.18
	Control	3.05	1.33	2.91	1.29	2.95	1.27
	Total	2.83	1.32	2.69	1.28	2.76	1.27
Controllability (1–5)	Group	4.03	0.33	4.05	0.49	3.97	0.47
	Individual	3.84	0.54	3.91	0.43	3.94	0.51
	Control	3.88	0.59	3.94	0.47	3.76	0.66
	Total	3.91	0.51	3.97	0.46	3.88	0.56
Coherence (1–5)	Group	3.25	0.52	3.69	0.57	3.67	0.47
	Individual	3.44	0.62	3.70	0.72	3.67	0.70
	Control	3.64	0.66	3.46	0.84	3.51	0.71
	Total	3.45	0.61	3.62	0.72	3.61	0.64
Emotional representations ^c^ (1–5)	Group	3.02	0.86	3.08	0.82	2.87	0.80
	Individual	2.31	1.14	2.33	1.03	2.32	1.18
	Control	3.37	0.57	2.89	0.79	3.11	0.59
	Total	2.90	0.98	2.75	0.93	2.76	0.94

^a^ Sample sizes are 16, 18, and 19 for the Group (G-EFT), Individual (I-EFT), and Control (CC) conditions, except for *n* = 18 at T2 for the control condition and *n* = 15 at T3 for the Group condition. ^b^ Or the equivalent time for the control condition. ^c^ The only significant difference at baseline was in emotional representations (F (2, 50) = 6.88, *p* = 0.002).

**Table 3 healthcare-13-01420-t003:** Satisfaction with the EFT instruction—three months after the intervention.

How Satisfied Are You with the Instruction on EFT?	Group Instruction (*n* = 15)	Individual Instruction (*n* = 18)	Total (*n* = 33)
Not at all	0	0	0
A little bit	0	1	1
Moderately	0	1	1
Quite a lot	6	5	11
Very much	9	11	20
Total	15	18	33

**Table 4 healthcare-13-01420-t004:** Frequency of practicing Emotional Freedom Technique (EFT).

	End of Intervention			Three Months Later		
Times a Week	Group Instruction	Individual Instruction	Total	Group Instruction	Individual Instruction	Total
0	4	3	7	5	8	13
1–2	1	6	7	2	3	5
3–5	6	6	12	8	7	15
6–7	5	3	8	0	0	0
Total	16	18	34	15	18	33

## Data Availability

The data presented in this study are available on request from the corresponding author due to concerns about patient privacy.

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
