# Peer review of "The Effectiveness of Group and Individual Training in Emotional Freedom Techniques for Patients in Remission from Melanoma: A Randomized Controlled Trial"

_healthcare, 2025, doi:10.3390/healthcare13121420_

Round 1
Reviewer 1 Report
Comments and Suggestions for Authors
1.About abstract, "Childhood and adult stress have been linked to the development 12 of several types of cancer". I wonder why does SCE stand out in the first sentence of the abstract?
2.About abstract, "in survivors of cutaneous melanoma", however, the study participants were not only survivors, please define survivor.
3.Small Sample Size and Power Issues
The study initially aimed for 57 participants (19 per group) but ended with 53, and attrition further reduced the sample (e.g., 15 in G-EFT at follow-up). While the authors note the study was powered to detect medium effects, many outcomes showed small effect sizes, raising questions about the reliability of non-significant findings (e.g., fear of recurrence, emotional representations). The post-hoc analysis of subgroups (e.g., participants who continued EFT practice) had even smaller samples, making conclusions speculative.
4.The study design did not mention blinding participants or practitioners, which introduces bias. Participants knew their group assignment (group vs. individual EFT vs. control), and the practitioner was aware of the intervention being delivered. This could influence self-reported outcomes like satisfaction and SUD scores.
5.The control group attended a lecture on stress and EFT before the study and completed questionnaires at the same intervals. This exposure might have influenced their behavior (e.g., seeking other stress-reduction methods), diluting differences between groups.
6.About self-Report Bias, most outcomes (e.g., wellbeing, fear of recurrence) relied on self-reported measures, which are prone to placebo effects, social desirability bias, or recall bias. The high satisfaction ratings (e.g., 97% recommending EFT) may reflect enthusiasm for the intervention rather than objective efficacy.
7.Participants were predominantly well-educated (74% held degrees), secular (89%), and in good health (mean self-rated health: 4/5). This limits applicability to broader populations, including those with lower socioeconomic status or higher baseline distress. The study excluded patients with psychiatric conditions or other cancers, despite stress interventions being particularly relevant for these groups.
8.Inconsistent Adherence to EFT Practice, despite high satisfaction, many participants stopped practicing EFT after the intervention (37% not practicing at 3 months). This undermines claims about EFT's long-term self-help benefits. The authors speculate about "booster sessions" but did not test this.
9.The use of one-tailed tests (p < 0.05) for directional hypotheses is unconventional and increases the risk of Type I errors. A two-tailed approach would be more conservative. The skin cancer risk score differed significantly between groups at baseline but was only controlled for in longitudinal analyses, not randomization. This suggests imperfect randomization.
Comments on the Quality of English LanguageMy native language is not English either. I won't make any comments.
Author Response
Response to Reviewer 1 Comments
|
||||||||||||||||
1. Summary |
|
|
||||||||||||||
Thank you very much for taking the time to review this manuscript. Please find the detailed responses below and the corresponding revisions/corrections highlighted in the re-submitted files.
|
||||||||||||||||
3. Point-by-point response to Comments and Suggestions for Authors
|
||||||||||||||||
Comment 1: About abstract, "Childhood and adult stress have been linked to the development of several types of cancer". I wonder why does SCE stand out in the first sentence of the abstract?
|
||||||||||||||||
Response 1: Thank you for pointing this out. We agree with this comment. Therefore, we replaced this sentence with the following one, which refers more directly to the background to this study’s aims: “A history of cancer has been linked to stress and concerns about recurrence”.
|
||||||||||||||||
Comment 2: About abstract, "in survivors of cutaneous melanoma", however, the study participants were not only survivors, please define survivor.
|
||||||||||||||||
Response 2: Due to space limitations in the abstract, we expanded the following sentence with additional brief information: “Fifty-three patients ages 18 and above diagnosed with melanoma (stage T1a–T2a) at least 6 months prior and not in active treatment.” (page 1, lines 17-18). In the Method section, Study population subsection, we added a more detailed description of the study population and defined survivor: “Melanoma staging for study participants was conducted based on Breslow tumor thickness, a standard measure of vertical tumor depth. Among the participants, 42.3% were diagnosed with melanoma in situ, 34.6% had thin melanomas (0.1–0.7 mm), and 23.0% had melanomas of intermediate thickness (0.8–1.2 mm). No participants presented with regional lymph node involvement or distant metastasis and none of the participants had received additional oncological treatments such as immunotherapy, targeted therapy, chemotherapy, or radiation. In the present study, we refer to the participants as melanoma survivors, i.e., as individuals who had completed active surgical treatment for melanoma and were in the follow-up phase of care.” (page 5, lines 246-254).
Comments 3: The study initially aimed for 57 participants (19 per group) but ended with 53, and attrition further reduced the sample (e.g., 15 in G-EFT at follow-up). While the authors note the study was powered to detect medium effects, many outcomes showed small effect sizes, raising questions about the reliability of non-significant findings (e.g., fear of recurrence, emotional representations). The post-hoc analysis of subgroups (e.g., participants who continued EFT practice) had even smaller samples, making conclusions speculative.
Response 3: The power analysis indicated that a sample of 45 (15 per condition) would provide a power of 0.90 to detect a medium-sized effect study. Taking into account the possibility of a 20% dropout rate, we initially aimed to recruit 57 participants. However, dropout was lower than we had estimated: We ended up with 53 participants who filled in the baseline questionnaires; two more had dropped out, one from the post-intervention follow-up and one from the 3-months follow-up, leaving a sample of 51 participants for the group by (all three) times analyses. However, as you pointed out, the actual effect sizes were smaller than we had expected. We are aware of the fact that this makes the conclusions speculative. We now modified the conclusion section of the manuscript (on page 21, lines 823-829), stating the EFT “may be effective in…” and that “The findings point to the need for further research on ways to optimize EFT training for this population and encourage adherence over time to increase its potential benefits.” Please also note that we mention the small sample size and effect sizes among the limitations and provide possible explanations for the modest effect sizes, some of which could guide future research in ways of minimizing these limitations. We also provide an analysis of all effect sizes, significant or not, for the full sample and for those who continued to practice EFT, which shows trends across time and measures (see Appendix A). This analysis generally supported the hypothesized direction of effects. Since this is the first such study on this population, we believe the findings provide indications of the ways in which EFT could be beneficial and they will encourage further research with larger samples. Note also that the limitations that are related to the characteristics of the study population led us to suggest that the effects found are an underestimate of the possible benefits of EFT for melanoma cancer survivors.
Comments 4: The study design did not mention blinding participants or practitioners, which introduces bias. Participants knew their group assignment (group vs. individual EFT vs. control), and the practitioner was aware of the intervention being delivered. This could influence self-reported outcomes like satisfaction and SUD scores.
Response 4: We agree and added to the limitations that “There was no way of blinding the study conditions to the participants or the practitioner, which could have introduced biases due to their expectations” (page 21, lines 790-791).
Comments 5: The control group attended a lecture on stress and EFT before the study and completed questionnaires at the same intervals. This exposure might have influenced their behavior (e.g., seeking other stress-reduction methods), diluting differences between groups.
Response 5: We totally agree and this is indicated in the limitations section: “…This may have raised their awareness of recurrence issues and led them to address their wellbeing in other ways, which might have diminished the differences between them and the intervention groups.” (page 21, lines 794-796). Unfortunately, there was no way of avoiding this because the control group had to be a waiting-list control group (not just people who only filled in questionnaires), so that the between-group comparisons would be within the same population of individuals who met the eligibility criteria and agreed to engage in EFT training. Providing information to all potential participants before consent was essential for recruitment and required by the Ethics committee (IRB).
Comments 6: About self-Report Bias, most outcomes (e.g., wellbeing, fear of recurrence) relied on self-reported measures, which are prone to placebo effects, social desirability bias, or recall bias. The high satisfaction ratings (e.g., 97% recommending EFT) may reflect enthusiasm for the intervention rather than objective efficacy.
Response 6: We agree and have added this to the study limitations: “Third, outcomes were assessed only with self-reported measures, because we aimed for changes in wellbeing, a subjective evaluation, yet such measures may be prone to various biases. SUD scores may also be biased by being collected in session by the instructor. However, all longitudinal outcome measures were collected confidentially online. Fourth, there was no way of blinding the study conditions to the participants or the practitioner, which could have introduced biases due to their expectations.” (page 21, lines 786-791).
Comments 7: Participants were predominantly well-educated (74% held degrees), secular (89%), and in good health (mean self-rated health: 4/5). This limits applicability to broader populations, including those with lower socioeconomic status or higher baseline distress. The study excluded patients with psychiatric conditions or other cancers, despite stress interventions being particularly relevant for these groups.
Response 7: We mentioned the relatively homogenous sample among the limitations (page 21, lines 778-780) and also noted that “…characteristics of the study population suggest that our findings underestimate the full potential of EFT for the general population of melanoma cancer survivors.” We agree that patients with other cancers may be particularly in need of a stress-reduction intervention. However, for research purposes, they had to be excluded because it is impossible to completely tease apart fear of recurrence of two or more types of cancer. Participants with psychiatric disorders were excluded due to safety, data validity, and ethical concerns. Serious psychiatric conditions may increase vulnerability to emotional destabilization during EFT sessions, affect adherence, and compromise data reliability. Additionally, such conditions or medication changes may confound outcomes, and impair the ability to provide informed consent.
Comments 8: Inconsistent Adherence to EFT Practice, despite high satisfaction, many participants stopped practicing EFT after the intervention (37% not practicing at 3 months). This undermines claims about EFT's long-term self-help benefits. The authors speculate about "booster sessions" but did not test this.
Response 8: Since this is a novel application of EFT training, we had no way of estimating adherence rates over the 3-month follow-up. We could only propose this for future studies, which we hope will be conducted, to test these and other possible expansions on the current protocol. The effect size computations shown in Appendix 1 show trends over time and measures, which are stronger among those who continued to practice. Therefore, we speculated there may be long-term benefits, which may require more investment in encouraging adherence. We added a final sentence to the conclusion paragraph along these lines: “The findings point to the need for further research on ways to optimize EFT training for this population and encourage adherence over time to increase its potential benefits.”
Comments 9: The use of one-tailed tests (p < 0.05) for directional hypotheses is unconventional and increases the risk of Type I errors. A two-tailed approach would be more conservative. The skin cancer risk score differed significantly between groups at baseline but was only controlled for in longitudinal analyses, not randomization. This suggests imperfect randomization.
Response 9: One-tailed tests are not common, but they are intended exactly for this type of study, in which the hypotheses state a clear direction of associations. This is why we decided to conduct all analyses as one-tailed tests. We provided the exact probability levels for all tests (not “p < 0.05” etc.) so the readers have access to the detailed information on significance levels (and in practice, all significant results would have also been significant at p > 0.05 in a two-tailed test). As for the difference in skin cancer risk scores at baseline: There was no way to prevent this because randomization was carried out after consent and the skin factor risk score was known only after participants had filled in the baseline questionnaire. Randomization is the most robust way to minimize the chance of pre-existing differences among groups but statistically, these could still occur, randomly, and it does not indicate imperfect randomization.
|
||||||||||||||||
4. Response to Comments on the Quality of English Language |
||||||||||||||||
Point 1: My native language is not English either. I won't make any comments.
|
||||||||||||||||
Response 1: We are not sure why this reviewer also marked at the beginning of the review that “The English could be improved to more clearly express the research”. A need for improving the language was not mentioned by the editor or the other two reviewers. All authors speak and write English at a very high level. One of the authors is a native English speaker and confirmed that the revised manuscript is accurate, scientifically and linguistically. |
||||||||||||||||
5. Additional clarifications None noted by the reviewer. We wish to thank you again for the careful reading of our manuscript and the helpful suggestions for clarification. We hope you will find the revised manuscript worthy of publication, and will be happy to address any remaining concerns. Many of the comments above refer to the limitations of the study and include suggestions for further emphasizing these limitations, which we did. We are aware of the limitations, yet we believe that being a novel study in this population, with a robust RCT design, its contribution is worthwhile: As we wrote above, we hope that it will encourage further research on this topic. We fully disclosed all significant and non-significant results and all limitations that we were aware of, assuming this will serve future research to identify fruitful aims to pursue when implementing and evaluating EFT interventions with cancer survivors, particularly melanoma survivors. |
||||||||||||||||
|
Reviewer 2 Report
Comments and Suggestions for Authors
Dear Authors,
Thank you for submitting your manuscript entitled “Randomized Controlled Trial of Emotional Freedom Techniques in Melanoma Survivors.” After a careful review, I offer the following specific observations and suggestions to strengthen the methodological quality and presentation of your work before potential acceptance.
1. Compliance with the CONSORT Statement
- Title and Abstract:
• Include initial capitals for the target population in the title (e.g., “Melanoma Survivors”).
• In the structured abstract (Background, Methods, Results, Conclusions), specify:
- Trial registration number and date (e.g., ClinicalTrials.gov NCT05421988; first registration: March 12, 2024).
- Eligibility criteria (age, melanoma stage, time since diagnosis).
- Randomization method (e.g., “block randomization with block size of 6, stratified by sex using software X”) and whether outcome assessors were blinded.
- Methods – Randomization and Blinding:
• Describe sequence generation (block size, stratification) and who prepared it.
• Detail allocation concealment (e.g., “sealed, opaque, numbered envelopes prepared by an independent member”).
• Clarify blinding: whether assessors and/or participants were blinded, or if the trial was open-label, with justification.
- Participant Flow:
• Add the total number of patients initially contacted and reasons for exclusion after screening in the CONSORT flow diagram.
• Specify if the analysis was intention-to-treat (ITT) or per-protocol, and report the final sample size per analysis.
- Sample Size:
• Provide parameters for the power calculation: expected effect size (e.g., d = 0.50), alpha level, desired power, number of groups and measures, and estimated dropout rate.
- Statistical Analysis:
• Include plans for handling missing data (imputation or available-case analysis).
• Report checks for normality and sphericity, and any corrections applied (e.g., Greenhouse–Geisser).
- Protocol and Transparency:
• Provide access to the full protocol and statistical analysis plan (e.g., as a supplement or in an online repository).
• Include the CONSORT 2010 checklist as an additional document.
2. Keywords (MeSH Terms)
Use valid MeSH descriptors (4–6 terms). For example:
Melanoma [Mesh]; Skin Neoplasms [Mesh]; Mind–Body Therapies [Mesh]; Stress, Psychological [Mesh]; Quality of Life [Mesh]; Cancer Survivors [Mesh]
3. Bibliography style (MDPI Format)
Please adopt the MDPI style consistently:
- Authors: up to six; if more, list the first six followed by “et al.”
- Article title: sentence case (capitalize only the first word and proper nouns).
- Journal name: abbreviated according to MEDLINE (italicized).
- Details: year; volume(issue):pages. DOI at the end.
Example:
Arnold M, Euvrard S, Boffetta P, Hémon B, Gondos A, Gavin A, et al. Incidence of melanoma in Europe: the EUROCARE-5 study. J. Eur. Acad. Dermatol. Venereol. 2022, 36(4), 495–503. https://doi.org/10.1111/jdv.18306
Ensure:
1. The journal title is correct for the publication year.
2. DOI format is accurate.
3. Abbreviations and punctuation are consistent.
4. Minor style and clarity points
- Avoid repetitive numerical data in the Introduction by citing melanoma incidence only once.
- Improve figure legends (Figures 2–5) by clearly stating the confidence interval (e.g., 95% CI) and defining abbreviations.
- Standardize footnote styles in tables (use superscripts “a”, “b” with corresponding notes).
Author Response
Response to Reviewer 2 Comments
|
||
1. Summary |
|
|
Thank you very much for taking the time to review this manuscript. Please find the detailed responses below and the corresponding revisions/corrections highlighted in the re-submitted files.
|
||
2. Questions for General Evaluation |
Reviewer’s Evaluation |
Response and Revisions |
Does the introduction provide sufficient background and include all relevant references? |
Yes/Can be improved/Must be improved/Not applicable |
Thank you for the positive general evaluation and for pointing out issues for improvement in the description of the methods and the results – please see detailed replies to the specific comments below. |
Is the research design appropriate? |
Yes/Can be improved/Must be improved/Not applicable |
|
Are the methods adequately described? |
Yes/Can be improved/Must be improved/Not applicable |
|
Are the results clearly presented? |
Yes/Can be improved/Must be improved/Not applicable |
|
Are the conclusions supported by the results? |
Yes/Can be improved/Must be improved/Not applicable |
|
3. Point-by-point response to Comments and Suggestions for Authors
Dear Authors, Thank you for the careful reading of the manuscript and for pointing out where the presentation of our study and its findings could be improved.
|
||
Comments 1: Compliance with the CONSORT Statement - Title and Abstract: Response: All main words in the title are capitalized, as instructed in the journal template. Response: The date was added (the rest of the information was already included in the abstract). Response: The missing information was added to the following sentence in the abstract: “Fifty-three patients ages 18 and above diagnosed with melanoma (stage T1a–T2a) at least 6 months prior and not in active treatment” (page 1, lines 17-18). In the methods section we now provide additional information regarding melanoma staging: “Melanoma staging for study participants was conducted based on Breslow tumor thickness, a standard measure of vertical tumor depth. Among the participants, 42.3% were diagnosed with melanoma in situ, 34.6% had thin melanomas (0.1–0.7 mm), and 23.0% had melanomas of intermediate thickness (0.8–1.2 mm). No participants presented with regional lymph node involvement or distant metastasis.” (page 5, lines 246-250). Response: We explained that randomization was conducted in one step after consent (and added a more detailed explanation in the methods section, see below). We added that the different outcome measures were obtained using online questionnaires; that the SUD scores were recorded by the EFT instructor, who was aware of the study condition.
- Methods – Randomization and Blinding: • Detail allocation concealment (e.g., “sealed, opaque, numbered envelopes prepared by an independent member”). • Clarify blinding: whether assessors and/or participants were blinded, or if the trial was open-label, with justification. Responses: Randomization was conducted in one step for the entire pool of consenting participants, after consent and before baseline data collection, so there was no need for blocks or stratification. We expanded the description in the Methods section: “Randomization to the study conditions was conducted in one step after consent and before the collection of baseline data. All fifty-six patients who consented to participate in the study were assigned code numbers and these were used in a computerized macro in Excel to randomize participants to the study conditions in a single step. Once randomized, the code numbers were converted back to the names to be invited to the sessions in each condition. Participants were assigned to one of the three study conditions, so that at least 18 participants would be included in each condition.” (page 7, lines 290-295).
- Participant Flow: Response: This information appears in the CONSORT diagram in Figure 1.
Response: Three participants dropped out before baseline data were collected (one was diagnosed with another type of cancer, one dropped out because his wife gave birth, and one consented and then declined, after being randomized). Since we did not have any baseline data for these participants, they could not be included in any ITT analyses. This information is described in full in the text on page 7 and concisely in the CONSORT diagram. Only two participants dropped out after baseline: One of the CC participants did not fill in the T2 questionnaires. This person’s data was included in the cross-sectional analyses and in the effect size computations in Appendix 1. One of the G-EFT participants did not fill in the T3 questionnaire and therefore was included only in the baseline cross-sectional analyses. Due to these small numbers, analyses were carried out with available data as is, i.e., per protocol, and not repeated with ITT. The CONSORT diagram and Table 2 clearly indicate the analysis sample in each group and the explanation above was added to the Statistical analysis section: “Projecting a 20% dropout rate, we aimed to recruit 57 participants (19 per condition) and first conduct analyses per protocol with all available data for each analysis. If drop-out rates were substantial, we planned to also conduct Intention-to-Treat (ITT) analyses with the full sample (using last available data point carried forward). As described above, we recruited and randomized 56 participants, and the baseline sample included 53 participants. Since drop-out rates over time were very low (one person in the CC did not fill in the questionnaire at T2 and one person in the G-EFT condition did not fill in the questionnaire at T3), we conducted analyses only per protocol: Cross-sectional comparisons between the study conditions and effect sizes for comparisons among study conditions in changes from beginning to end of study were conducted with data from 52 participants and comparisons across all three time points were carried out on the final analysis sample of 51 participants. Actual power for detecting a medium-sized effect was 0.95.” (page 11, lines 478-490).
- Sample Size: Response: We added d and alpha (page 11, line 478).
- Statistical Analysis: Response: We expanded the statistical analyses section and now include the following information (in addition to the paragraph pasted above): “Among the participants who filled in each questionnaire, there was no construct-level missingness.” (page 12, lines 491-492). And: “We considered Mauchly’s test of sphericity, for which a significant result means that the assumptions on which this mixed-model approach is based were violated. The test yielded non-significant results for all but two measures: Happiness and spiritual well-being. In both cases, the Greenhouse-Geisser epsilon value was > .80. For these two measures, sphericity was not assumed and we used the Huynh-Feldt adjustment (which is recommended when the Greenhouse-Geisser epsilon value is > .75; in practice we also computed the findings with the Greenhouse-Geisser adjustment and the results were very similar).” (page 12, lines 501-508).
- Protocol and Transparency: Response: We expanded the study procedure and the statistical analysis sections so that all information is transparent and included in the manuscript. Therefore, we did not upload additional documents, as they be redundant and would not add information for the readers. The EFT intervention protocol is available online (cited as reference #26).
Response: We apologize for this accidental omission in the initial submission. We will upload it with the revised submission.
|
||
Comments 2: Use valid MeSH descriptors (4–6 terms). For example: Response 2: The journal instructions ask for three to ten keywords, with no mention of MeSH descriptors. In correspondence with the comment here, we adapted the list of keywords so that it now includes only MeSH, with one exception, which seems essential for the current manuscript: Emotional Freedom Techniques. The list of keywords now includes: ”Melanoma; Skin Neoplasms; Cancer Survivors; Mind–Body Therapies; Psychological Wellbeing; Stress, Psychological; Emotional Freedom Techniques.” (page 1, lines 32-33).
|
||
Comments 3: Bibliography style (MDPI Format) Please adopt the MDPI style consistently:
Response 3: The references were entered with ENDNOTE using the MDPI output style, which truncated authors lists using “et al.” after ten authors. Following the comment above, we removed authors seven and on in all such references. We checked that article titles are properly capitalized, journal names are abbreviated, and DOI appears at the end. One reference was accidentally not entered through ENDNOTE and this was corrected (it is now reference #94 and the numbering from there on was moved accordingly).
Comments 4: - Avoid repetitive numerical data in the Introduction by citing melanoma incidence only once. Response 4: Thank you for noticing these issues. We corrected them as suggested in the comment above. The figure captions were also expanded following a suggestion from Reviewer 3.
|
||
4. Response to Comments on the Quality of English Language |
||
(There were no comments on the quality of English language). |
||
|
||
5. Additional clarifications |
||
|
Reviewer 3 Report
Comments and Suggestions for Authors
I recommend acceptance with minor editorial revisions.
Figures – Enhance Interpretability
Figures 2 through 5 could benefit from clearer labeling and legends. For example: Clarify group labels (G-EFT, I-EFT, CC) directly in figure captions. Indicate whether differences shown are statistically significant in the figure notes (e.g., “*p < 0.05” where relevant).
Language/Stylistic Corrections
While the manuscript is overall well-written, a few sentences throughout would benefit from minor rephrasing for smoother flow. Examples include: Abstract, line 25: “Statistically nonsignificant effects were found…” → consider rephrasing to “No statistically significant effects were found…” for improved clarity. Discussion, line 676: “...reported lower happiness at the end of the intervention and then an increase.” → consider “...reported a temporary decrease in happiness post-intervention, followed by a subsequent increase.” Consistency in Terminology Ensure consistency in the naming of the intervention conditions. For instance, sometimes "G-EFT" and "I-EFT" are used, and in other places, "group EFT" or "individual instruction." Standardizing this throughout the manuscript would enhance clarity.
Conclusion Paragraph – Strengthen Final Sentence: The conclusion could end with a stronger closing sentence reinforcing the practical relevance or potential for future implementation of EFT. Currently, it ends somewhat abruptly.
Author Response
Response to Reviewer 3 Comments
|
||
1. Summary |
|
|
Thank you very much for taking the time to review this manuscript. Please find our detailed responses below and the corresponding revisions/corrections highlighted in the re-submitted files.
|
||
2. Questions for General Evaluation |
Reviewer’s Evaluation |
Response and Revisions |
Does the introduction provide sufficient background and include all relevant references? |
Yes/Can be improved/Must be improved/Not applicable |
We appreciate the positive general evaluation. Replies to specific comments appear below. |
Is the research design appropriate? |
Yes/Can be improved/Must be improved/Not applicable |
|
Are the methods adequately described? |
Yes/Can be improved/Must be improved/Not applicable |
|
Are the results clearly presented? |
Yes/Can be improved/Must be improved/Not applicable |
|
Are the conclusions supported by the results? |
Yes/Can be improved/Must be improved/Not applicable |
|
I recommend acceptance with minor editorial revisions.
3. Point-by-point response to Comments and Suggestions for Authors
|
||
Comments 1: Figures – Enhance Interpretability
|
||
Response 1: Thank you for pointing this out. You are absolutely right that figures should be legible on their own. Following this comment, we realized that more information was needed in the figure captions: We added the group labels and indicated the significant finding. Since the findings mostly involved significant interactions, we could not mark them within the figure but as suggested, we indicated in the figure caption what was significant and at what probability level. For example, the Figure 2 caption now reads (page 14, lines 560-565): Figure 2. Levels of coherence (understanding prevention of recurrence) and 90% confidence intervals (CIs) across time in the three conditions: Group EFT training, Individual EFT training, and Control condition (controlling for skin cancer risk scores). The Figure presents a significant linear and quadratic study condition X time interaction, i.e., a significant difference (p < .05) between the pattern of improvement and then stability in coherence levels in the two EFT conditions and the more stable scores over time in the Control condition.
|
||
Comments 2: Language/Stylistic Corrections
|
||
Response 2: Thank you for the helpful suggestions, which we adopted in full. We now use G-EFT and I-EFT consistently in the text and have changed “group” to “condition” (or “study condition”, where appropriate) all along in order to prevent any confusion between the Group versus Individual EFT training and the study conditions. In the figures we used the full terms (Group EFT, Individual EFT). |
||
Comments 3: Conclusion Paragraph – Strengthen Final Sentence: The conclusion could end with a stronger closing sentence reinforcing the practical relevance or potential for future implementation of EFT. Currently, it ends somewhat abruptly.
Response 3: Thank you for this suggestion. This is the current conclusion paragraph, with an added final sentence: “Our study offers evidence that teaching EFT to melanoma patients as a non-pharmacological and non-invasive self-help method is well-received and may be effective in reducing psychological stress, raising patient awareness of possible links between traumatic emotional events and well-being, improving spiritual wellbeing and the understanding of measures that reduce the risk of cancer recurrence. The findings point to the need for further research on ways to optimize EFT training for this population and encourage adherence over time to increase its potential benefits.” (pages 21-22, lines 823-829).
4. Response to Comments on the Quality of English Language |
||
Point 1: |
||
Response 1: There were no comments in this section. |
||
5. Additional clarifications |
||
None. |